# Effect of Biochar Diet Supplementation on Chicken Broilers Performance, NH_3_ and Odor Emissions and Meat Consumer Acceptance

**DOI:** 10.3390/ani10091539

**Published:** 2020-09-01

**Authors:** Kajetan Kalus, Damian Konkol, Mariusz Korczyński, Jacek A. Koziel, Sebastian Opaliński

**Affiliations:** 1Department of Environment Hygiene and Animal Welfare, Wroclaw University of Environmental and Life Sciences, 51-630 Wrocław, Poland; damian.konkol@upwr.edu.pl (D.K.); mariusz.korczynski@upwr.edu.pl (M.K.); sebastian.opalinski@upwr.edu.pl (S.O.); 2Department of Agricultural and Biosystems Engineering, Iowa State University, Ames, IA 50011, USA; koziel@iastate.edu

**Keywords:** poultry, sustainability, air quality, mitigation, ammonia, olfactometry, manure, feed additives, biocoal, environmental analysis

## Abstract

**Simple Summary:**

Poland leads the EU in poultry meat production. Poultry diet supplementation is actively researched to improve the sustainability of the industry and to lower the environmental footprint. We tested a hypothesis if biochar (a carbon-rich material) addition to the diet could address selected environmental goals without compromising production parameters and consumer preferences. The results show that supplementation of chicken broilers diet with biochar contributed to the reduction of ammonia emissions from manure but increased feed conversion ratio. The average body weight and daily weight gain were lower. Meat consumer acceptance was not influenced. In general, the use of biochar as a feed additive could be beneficial to reduce the emissions of ammonia (and potentially odor) from manure, but it worsens some of the key production parameters.

**Abstract:**

The aim of this research was to evaluate the effect of biochar diet supplementation for broiler chickens on (1) ammonia and odor emissions from manure, (2) feed conversion ratio and daily weight gain, and (3) selected meat quality and sensory parameters. Beechwood biochar (BC, 2 and 4%) and BC–glycerin–aluminosilicates mix (BCM, 3 and 6%) were tested as dietary additives. A total of 750 chicken broilers (Ross 308) were divided into five dietary groups with five replicates per group (*n* = 5, 30 birds in each replicate) and reared on a littered floor for 5 weeks. Both feed additives showed a significant reduction of ammonia emissions by up to 17%, while the reduction of odor emissions was not statistically significant. The feed conversion ratio increased by 8% for the highest concentration of the mixture. The change of the treated broilers’ average body weight ranged in the last week of the experiment from 0 to −7%, with the most negative effect for the highest dose of the mixture. Sensory analysis of the sous-vide cooked breasts showed no significant differences.

## 1. Introduction

Poland is the biggest European Union (EU) poultry meat producer. Over 2.5 million tonnes of meat was produced in 2019, an equivalent of ~17% of the total EU production [1]. Approximately 83% of the meat comes from chicken broilers. Intensive poultry production raises concerns about gaseous emissions (mainly ammonia, NH_3_) and its impact on workers, birds, and the environment. Additionally, odor nuisance can be a significant problem for nearby communities [2]. Mitigation of the gaseous emissions and barn indoor air quality are serious challenges for regulatory agencies and farmers.

Most of the technologies for mitigating gaseous emissions from livestock were tested for swine [3,4,5]. Those technologies include air cleaning systems, manure additives, or dietary manipulation [6,7,8]. Biochar is a relatively new topic of scientific research and could be utilized as a feed additive. Results of dietary supplementation with biochar are still few, but some are promising, i.e., in terms of performance parameters or mitigation of greenhouse gas emissions [9,10].

Biochar is obtained via decomposition of biomass at high temperatures and no oxygen environment (via pyrolysis or torrefaction), where organic material yields a solid carbonaceous powdery material and gaseous products. The solid residue is termed “biochar” to indicate its biomass origin, e.g., from agricultural production or as a by-product of biorenewable energy production. The physicochemical properties of biochar depend on feedstock material and the thermochemical process [11]. It is not surprising, then, that biochar addition to poultry diet can result in a wide range of outcomes. For example, biochar diet inclusion resulted in an increase in feed conversion ratio (FCR) and a decrease in body weight gain of chicken broilers [12,13]. Others have reported an improvement of production parameters (higher egg production and egg weight) and FCR for laying hens [14,15].

We have recently reported on the effects of beechwood biochar (BC) and BC–aluminosilicates–glycerin mixture (BCM) supplementation of laying hens diet [16]. The BC addition resulted in an increase in daily feed intake, while BCM reduced it. The treatments increased laying performance by 6%, shell resistance to crushing by up to 10%, and shell thickness by up to 6%. The sensory analysis of hardboiled eggs showed no significant differences between treatments. The total N content in manure was lower by 4 to 20%; its pH numerically increased [16]. The majority of treatments had a positive (however not statistically significant) effect on NH_3_ emissions mitigation. The mitigation of volatile organic compound (VOC) emissions was also not statistically significant. Thus, these first series of experiments with BC and BCM treatments and laying hens point out to the possibility of simultaneously reducing the environmental impact while improving production parameters in other poultry production systems.

In this research, we evaluated the influence of BC and BCM in the broiler chickens diet. BC has been used as a feed additive, and BCM was also tested for comparison, the latter being an example of marketed BC-based feed additive. Birds’ performance and emissions from manure were evaluated, including FCR, average daily weight gain (ADWG), average breasts and drumsticks mass, and their ratio to the carcass’ mass together with NH_3_ and odor emissions from the manure. Additionally, for the first time for BC-based dietary treatment, sensory analysis of sous-vide cooked chicken breasts had been conducted for their visual appearance, color, smell, texture, and taste. Our working hypothesis was that the BC and BCM diet supplementation mitigates NH_3_ and odor emissions from manure without negatively impacting broilers’ production parameters and meat quality.

## 2. Materials and Methods

### 2.1. Ethical Approval

This study adhered to the Polish law (Act of 15.01.2015 on the Protection of Animals Used for Scientific and Educational Purposes), which pertains to the protection of animals used for scientific or educational purposes. Details are presented elsewhere [16]. In short, the experiment was not harmful to the broilers relative to the abovementioned law, and ethical approval was not required.

### 2.2. Birds, Experimental Design and Housing

A total of 750 one-day-old male, Ross 308 chickens (Malec Hatchery Ltd., Góra Kalwaria, Poland), were allotted randomly to 1 of 5 feed diet treatments. The broilers were housed on a littered floor (wheat straw), 15 birds per m^2^, with ad libitum access to feed and water (Figure A1, Appendix A) for 35 days, in accordance with European Commission Regulation (EC) no 429/2008 of 25 April 2008 on applications, assessment, and authorisation of feed additives. The setpoint for temperature was 32 °C for day 1; after that, it was slowly reduced to 22 °C till day 21 and remained constant until day 35. The 10 min of light was provided every 6 h during the first week and then for 18 h every 6 h, except for the last 3 days when the light was provided for 24 h.

### 2.3. Feed Additives

Two additives: BC and BCM consisting of 67% of the BC, 24% of aluminosilicates (an anti-caking agent), and 9% of glycerin (an anti-dusting agent) (Figure A2, Appendix A) were investigated. A detailed summary of the properties of the BC and aluminosilicates are presented elsewhere (Table 1 in [16]). Briefly, the BC pH was 9.0, 350 g×dm^−3^ bulk density, <8% moisture, and <10% ash content, respectively. The aluminosilicates had a pH of 8.7, 741 g×dm^−3^ bulk density, and <6.2% moisture content.

Feed was prepared in a small-scale mechanical mixer every 3 days, separately for each group, to ensure homogeneity of the treatments. All chicken broilers were fed the same (basal) diet. The diet was formulated using nutrient recommendations for broilers (Table 1). The control (C) group was fed the basal diet only. The BC2 and BC4 treatment groups were fed with the 2% and 4% addition of BC by mass, respectively. The BCM2 and BCM4 treatment groups were fed with the 3% and 6% addition of BCM by mass, respectively (Figure 1). The BC mass in the BCM2 and BCM4 groups was numerically identical to the BC2 and BC4 groups, respectively. The treatment doses were based on the results from our recent study [16] and other research in a similar field [12,13,17,18,19].

### 2.4. Production Parameters, Meat and Carcass Traits

The average body weight (ABW) together with an average daily weight gain (ADWG) were estimated once a week. Feed conversion ratio (FCR) was estimated at the end of the study by taking the average mass of the consumed feed, reduced by the mass of the feed that remained in the feeder and dividing it by the average mass of the broiler, reduced by its initial mass. Average daily feed intake (ADFI) was calculated as a product of FCR and ABW. A total of 40 carcasses (8 per treatment) were analyzed for breasts and drumsticks masses, and their mass ratio to the carcass’s mass. Evaluation of the BC and BCM effectiveness on ABW, ADWG, and meat parameters was made using a percentage in relation to the control (PRC, %) calculated as the ratio between treatments and the control parameters’ means.

Additionally, sensory analysis of the chicken breasts was conducted at the end of the trial. Seventy-five panelists were randomly selected from among the university employees and students in order to evaluate the consumers’ acceptance of the meat’s appearance, color, smell, texture, and taste (Figure A3, Appendix A). Sensory analyses were conducted using a completely randomized design. The panelists were presented with sous-vide cooked chicken breast samples and were asked to rate each parameter on the 1-to-5 scale (worst-to-best).

### 2.5. Manure Properties

#### 2.5.1. Manure Sampling for NH_3_ Analysis

Chicken broilers’ manure was collected during the last week of the experiment. The birds were manually moved for 1 h into large plastic enclosures, separately assigned to each pen. Then the broilers were moved back to the pens, and 200 g of the representative manure samples were collected from the containers, separately for each replication. The amount of collected manure (200 g) related to the surface area of laboratory glass treatment containers corresponds to the amount of manure produced from 1 m^2^ of a standard floor-raised poultry barn over 1 week (Figure A4, Appendix A). The manure samples were transported to the laboratory immediately after the collection for measurements of NH_3_ emissions.

#### 2.5.2. NH_3_ Emissions

The effects of BC and BCM dietary treatments were evaluated similarly to the protocol described elsewhere [16]. Briefly, manure samples were placed in special containers, and headspace was sampled by aspirators through impingers with acid. The NH_3_ concentration was determined using the standard method (PN-71/Z-04041). The mitigation effect was evaluated using PRC, (%) calculated as the ratio between treatments and the control mean (of 5 replicates) NH_3_ concentrations.

#### 2.5.3. Odor Emission Analysis

During the third week of the experiment, an olfactometry test has been conducted in order to compare odor concentration (control group vs. treatment groups) with the highest doses of additives (BC4 and BCM4). Only the higher doses of additives were used to evaluate if the treatments mitigate odorous emissions at all, as a screening experiment. The samples were taken according to VDI 3880:2011 and PN-EN 13725:2007 standards [20]. A steel hood, 0.5 m^2^ of surface area, was put directly onto the investigated groups’ pen floors, and the headspace air was pulled into PET bags for 20 min using a vacuum pump (Figure A5, Appendix A). The samples were transported immediately after collection to the accredited Olfactometric Laboratory (Wroclaw University of Science and Technology) for further analysis with the use of ECOMA TO8 olfactometer (Olfasense, Kiel, Germany) by four panelists. The mitigation effect was evaluated using PRC (%) calculated as the ratio between treatments and the control mean (of 3 replicates) European odor units.

### 2.6. Statistical Analyses

The experiment was conducted as a completely randomized design with 5 treatment groups; each treatment replicated 5 times (30 birds/replicate). The protocols and tests are described in detail elsewhere [16]. Briefly, all the reported data are mean values that were tested for normality with the Shapiro–Wilk test. A one-way analysis of variance was used for normal distributions. The Kruskal–Wallis test was used for distributions that were not normal. Statistical significance was assigned at *p*-value < 0.05.

## 3. Results

### 3.1. Production and Meat Parameters

A comparison between the groups of ABW, ADWG, FCR, and ADFI together with PRC, is presented in Table 2. ABW and ADWG are compared on a weekly basis, and for those parameters, PRC is presented only for the last week of the experiment. As early as after the first week of the experiment, the treatments already showed a negative effect on the ABW, with the highest and statistically significant decrease in the BCM4 group. No statistically significant differences were observed during the second week, yet the differences started to decrease, especially in groups supplemented with the lower concentration of investigated additives (groups BC2 and BCM2). In the third week, differences in ABW between the control and treatment groups with lower concentrations (BC2, BCM2) were lower but statistically insignificant, while higher concentration groups (BC4, BCM4) showed significantly lower values. The fourth week showed no significant differences. After the final week, the BCM4 group showed a 7% lower average body weight compared to the control group.

Changes in ADWG were also observable but statistically insignificant, with an only significant difference after the first week, with a 12% difference between control and BCM4 groups (Table 2). The overall effect of dietary inclusion on FCR was negative with FCR compared to the control increased significantly by 8% for the BCM4 group. ADFI was not statistically different between any of the investigated groups.

Results for drumsticks and breast masses to the carcass mass ratio are presented in Table 3. As the used feed additives contributed to the reduction of birds’ body weight, drumsticks and breast masses were lower as well, with also lower drumstick/carcass and breasts/carcass mass ratios; however, the differences are not statistically significant.

Panelists’ mean grades for sous-vide cooked chicken breasts overall appearance, color, smell, texture, and taste ranged from 3.06 to 3.96, and are presented in Table 4. There were no statistically significant differences between all of the investigated broilers’ meat samples.

### 3.2. Ammonia

Mean NH_3_ concentrations values, along with PRC, are presented in Table 5. NH_3_ concentrations measured in all of the treated manure’s headspace were lower (and statistically significant) by 15, 14, 17, and 15% for BC2, BC4, BCM2, and BCM4 groups, respectively, compared with the control group. The biochar concentration and type of used additive seem to be of less importance, as all of the investigated additives showed comparable reductions of NH_3_ emission. Biochar addition to chicken broilers’ diet has a positive effect in reducing NH_3_ emissions from the manure.

### 3.3. Odors

Mean European odor units (OU_E_ × m^−3^) measured for the litter’s headspace are presented in Table 6. The odor concentration was lower in the BC4 and BCM4 groups by 32 and 37%, respectively, compared with the control group. However, the differences were not statistically significant. Due to the limitations of the study, only the highest concentrations of used additives were included in odor emission assessment, and sampling was performed only during the third week of the experiment. The lack of statistical significance most likely results from the low amount of replicates and only one sampling day. However, while more replications are recommended for confirmation, a general assumption can be made that the biochar addition to broiler chickens’ diet can potentially reduce odorous emissions from the manure.

## 4. Discussion

The main aim was to investigate the use of biochar (BC2 and BC4 groups) and market-available biochar-based feed additive (BCM2 and BCM4 groups) as a supplement of chicken broilers diet, in order to assess the reduction potential of NH_3_ and odor emissions which are generated from poultry meat production sector. For practical application, such dietary manipulation must not influence the production parameters of the birds adversely. While the investigated feed additives contributed to lower NH_3_ (and potentially odor) emissions, the production parameters were worsened for the BCM4 group.

Additionally, for the first time for biochar supplementation of broiler chickens’ diet, consumers’ acceptance of the sous-vide cooked chicken breasts was evaluated. No statistically significant differences were found between any of the treated/untreated samples.

Lower concentration of the feed mixture (3%, BCM2 group) performed the best from all treatment groups, and compared to the control there was no significant difference in broilers’ average body weight, and their ADWG was better during the last 2 weeks of the experiment, without significant influence on the FCR.

The effect of NH_3_ and odor emission reduction could be explained by the adsorption of the gases on the surface of the biochar; biochar could also alter the microbial activity in the birds’ gut leading to lower production of NH_3_ and odorous VOCs.

The drop in production performance of the BCM4 group is likely the effect of lower availability of nutrients due to the highest concentration of biochar that has been used in that treatment group. Only BCM2 showed a positive effect on production parameters that could be explained by compensation for the lack of nutrients with the effect of glycerin acting as an energy source.

Kutlu et al. (2001) [17] reported a positive effect of 2.5, 5.0 and 10% of oak charcoal additions to the chicken broilers feed, where, although the results were not statistically significant, 2.5% concentration had the numerically best impact with a 2% better body weight gain (BWG), 1% lower FCR and only 1% higher feed intake. Higher concentrations did not have a positive effect on the broilers’ production parameters. The second experiment conducted in the research, where charcoal was added to starter and finisher diets, showed that groups receiving the treatment have higher BWG and carcass weight than control, with 2.5% addition of charcoal only to the starter diet having the best effect of 8% higher carcass weight, 8% higher BWG and 3% lower FCR. It is interesting that the abundant addition of charcoal in the amount of 10% did not contribute to a severe drop in the production parameters of the birds. However, differences between the abovementioned results and the present study may result from the different types of biocoal used (oak charcoal vs. beech biochar) and the fact that broilers have constantly been provided with light.

Kana et al. (2011) [18] used canarium seed and maize cob charcoals in the considerably lower amount of 0.2, 0.4, and 0.6%. Both charcoals significantly improved body weight and BWG and decreased feed intake, but FCR did not differ significantly due to the treatments. It is worth noting that FCR, on average, was 0.47 higher than FCR in the present study. The best feed conversion ratio was obtained with the use of 0.6% maize cob-based charcoal. In the case of the concentrations higher than 0.6% of any used charcoal, a slight decrease in mean feed consumption, average final weight, and average weight gain were observed. It was suggested that, in the case of production parameters, 2% is a limiting concentration of biochar in the feed, which supports the results of the present study. Kana et al.’s (2011) study does not provide any data on odors or NH_3_ emission from the treated broilers’ manure.

Evans et al. (2015) suggested that high concentrations of biochar in the feed can contribute to the worsening of the production parameters of the birds [12]. The researchers observed higher feed intake (8%), lower weight gain (2%), higher FCR (11%), and higher mortality among broilers fed with 7% amendment of poultry-litter derived biochar. Such effect was explained by high arsenic content in the biochar (99 ppm). A follow-up research (2017) [13] with 2 and 4% amendment of the same biochar, but with reduced toxic metals content, resulted in a significant 7% increase of FCR and 9% decrease of live weight gain for 2 and 4% of the feed amendment, respectively. The results confirm the negative effect of biochar on the production parameters of the broilers.

Prasai et al. (2018) investigated green waste biochar addition (1, 2, and 4%) to broilers’ diet and its effect on manure properties, including NH_3_ emission from the manure [19]. The 2 and 4% treatments lowered manure N content significantly by 17% and 27%, respectively. The N was likely lost via gaseous NH_3_, as emissions of NH_3_ were (contrary to the present study) higher by 47 and 43%, respectively. The difference could possibly be caused by the different source materials and processes used to obtain biochar, which could be characterized by different surface areas responsible for the adsorption of NH_3_. Furthermore, different biochars could influence microbial activity in the birds’ gut differently and, e.g., promote NH_3_ generation.

As the application of biochar for broilers’ feed is a relatively new field of study, there is little research on the subject to compare and discuss with, especially when it comes to evaluation of influence on gaseous emissions from the manure and consumer preferences. To date, only one piece of research on NH_3_ emission from biochar-fed broilers’ manure is available, and the data on the emission contradict the present study. A comparison to other research focused on the production parameters is inconclusive. Some authors report an overall positive effect on production parameters, even with relatively high concentrations of biochar used. In contrast, others report worsening outcomes for biochar concentrations lower than used in the present study.

## 5. Conclusions

In the present study, the biochar and the biochar-based feed additive were used as an inclusion to chicken broilers’ diet. The effect of such supplementation was overall positive in terms of reduction of NH_3_ (and potentially odor) emissions while the meat’s consumer preference was not altered. However, production parameters of the birds were worsened, with a statistically significant negative effect for the highest dose of biochar–glycerin–aluminosilicates mixture (BCM4) used. Based on the results from the present study, only BCM with the lowest investigated biochar dose could be recommended for a potential, effective dietary inclusion for chicken broilers. It should be noted that many factors can influence the physicochemical properties of biochar, starting with the biochar feedstock material and the thermochemical process. Thus, it is recommended that more research emphasis is placed on the proper functionalizing of biochar aimed to comprehensively address environmental, production, and consumer concerns.

## Figures and Tables

**Figure 1 animals-10-01539-f001:**
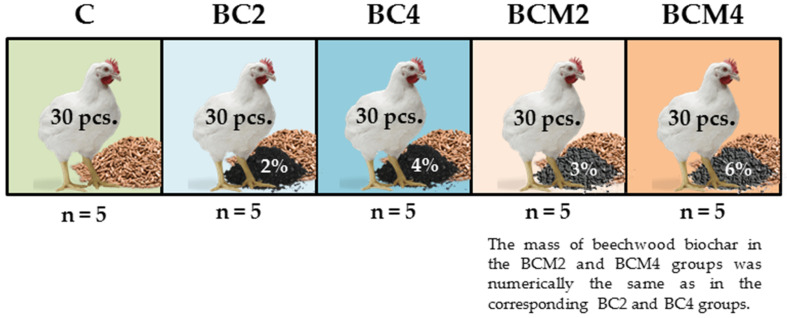
Visual representation of the experimental design.

**Table 1 animals-10-01539-t001:** Basal diet—chemical composition.

Ingredient	Content (%)		
	Starter (0–7 Days)	Grower (8–26 Days)	Finisher (5–27 Days)
Corn	32.03	36.18	35.22
Wheat	29.10	29.50	32.10
Post-extraction soybean meal	27.30	21.30	18.20
Lard	-	3.00	3.10
Sunflower cake	2.00	2.50	3.00
Fish meal	2.00	-	-
Post-extraction rapeseed meal	1.60	0.90	3.00
Post-extraction sunflower seed meal	-	1.50	0.60
Medium-chain fatty acids	1.50	1.20	1.20
Limestone	1.17	0.73	0.65
Guar meal	-	1.00	1.20
Soybean oil	0.80	-	-
Monocalcium phosphate	0.59	-	-
Dicalcium phosphate	0.40	0.70	0.36
Sodium chloride	0.22	0.21	0.21
L-Lysine sulphate	0.20	0.20	0.20
Mineral-vitamin premix ^1^	0.20	0.20	0.20
L-Methionine	0.15	0.24	0.21
Rhodimet 88	0.15	-	-
L-Threonine	0.12	0.11	0.10
Sodium bicarbonate	0.12	0.12	0.12
L-Lysine	0.09	0.17	0.16
Choline chloride	0.07	0.07	0.06
Mycofix Select 5.E	0.06	0.05	0.05
Sacox 120	0.06	0.06	-
L-Valine	0.04	0.04	0.04
Phytase	0.02	0.01	0.01
Xylanase	0.01	0.01	0.01

^1^ Contributes (mg × kg^−1^ diet): Co, 1 (as CoCO_3_); Cu, 9 (as CuSO_4_∙5H_2_O); Fe, 30 (as FeSO_4_∙H_2_O); Mn, 80 (as MnO_2_); Se, 0.4 (as a_2_SeO_3_∙5H_2_O); Zn, 80 (as ZnO); butylated hydroxytoluene, 0.6; butylated hydroxyamisole, 0.06; etoxyquin, 0.1.

**Table 2 animals-10-01539-t002:** Comparison of biochar-fed broilers’ production parameters.

	Broilers’ Production Parameters (*n* = 5)							
	Week	C	BC2	BC4	BCM2	BCM4	SEM	*p*-Value
**ABW (g)**	1	148.33 ^a^	140.58 ^ab^	138.33 ^ab^	142.98 ^ab^	135.12 ^b^	1.49	0.0466
	2	376.56	374.45	344.57	364.72	364.81	3.32	0.161
	3	775.23 ^a^	777.02 ^a^	733.79 ^b^	757.56 ^ab^	738.13 ^b^	5.74	0.0329
	4	1304.49	1315.59	1263.73	1305.31	1244.52	11.36	0.108
	5	1824.14 ^a^	1794.70 ^ab^	1758.65 ^ab^	1820.83 ^ab^	1701.73 ^b^	12.19	0.0059
	PRC (%)		−2	−4	−1	−7		
**ADWG (g)**	1	15.47 ^a^	14.36 ^ab^	14.12 ^ab^	14.70 ^ab^	13.58 ^b^	0.21	0.0467
	2	37.74	33.38	31.08	34.67	35.40	1.20	0.232
	3	56.79	58.15	54.31	53.45	53.44	0.71	0.119
	4	75.61	78.57	78.88	84.94	74.17	1.64	0.214
	5	72.90	69.54	71.47	75.91	65.97	1.51	0.278
	PRC (%)		−5	−2	+4	−10		
**FCR**		1.8 ^a^	1.83 ^ab^	1.87 ^ab^	1.81 ^a^	1.94 ^b^	0.01	0.0047
	PRC (%)		+2	+4	+1	+8		
**ADFI (g)**		93.81	93.84	93.96	94.16	94.33	0.611	0.545
	PRC (%)		+0.03	+0.16	+0.37	+0.55		

ABW = average body weight; ADWG = average daily weight gain; FCR = feed conversion ratio; ADFI = average daily feed intake; C = control group; BC2 = 2% of beechwood biochar by mass; BC4 = 4% of beechwood biochar by mass; BCM2 = 3% of beechwood biochar-based additive by mass; BCM4 = 6% beechwood biochar-based additive by mass. The beechwood biochar mass in the BCM2 and BCM4 groups was identical to BC2 and BC4 groups, respectively. ^a,b^ Mean values in the same row marked with different superscript indicate statistical significance (*p* < 0.05). Mean values with no superscript are not significantly different from any other values. PRC = percentage in relation to the control. Negative PRC indicates a reduction, while positive PRC values indicate an increase, in comparison to the control group.

**Table 3 animals-10-01539-t003:** Drumstick and breast mass and their ratio to carcass mass.

	Drumsticks and Breasts Values (*n* = 8)						
	C	BC2	BC4	BCM2	BCM4	SEM	*p*-Value
Drumstick to carcass mass ratio (%)	8.82	8.63	8.67	8.84	9.07	0.08	0.479
	PRC (%)	−2	−2	0	+3		
Breasts to carcass mass ratio (%)	18.04	17.32	16.50	17.08	16.77	0.23	0.454
	PRC (%)	−4	−9	−5	−7		

C = control group; BC2 = 2% of beechwood biochar by mass; BC4 = 4% of beechwood biochar by mass; BCM2 = 3% of beechwood biochar-based additive by mass; BCM4 = 6% beechwood biochar-based additive by mass. The beechwood biochar mass in the BCM2 and BCM4 groups was identical to BC2 and BC4 groups, respectively. PRC = percentage in relation to the control. Negative PRC indicates a reduction, while positive PRC values indicate an increase, in comparison to the control group.

**Table 4 animals-10-01539-t004:** Sensory traits of sous-vide cooked chicken breasts after biochar-based treatment.

	Meat’s Sensory Parameters (*n* = 75)						
	C	BC2	BC4	BCM2	BCM4	SEM	*p*-Value
Appearance	3.85	3.87	3.75	3.89	3.81	0.05	0.905
Color	3.96	3.88	3.74	3.88	3.81	0.04	0.422
Smell	3.66	3.69	3.74	3.81	3.77	0.04	0.844
Texture	3.23	3.54	3.24	3.20	3.06	0.05	0.222
Taste	3.31	3.45	3.27	3.41	3.20	0.05	0.603

C = control group; BC2 = 2% of beechwood biochar by mass; BC4 = 4% of beechwood biochar by mass; BCM2 = 3% of beechwood biochar-based additive by mass; BCM4 = 6% beechwood biochar-based additive by mass. The beechwood biochar mass in the BCM2 and BCM4 groups was identical to BC2 and BC4 groups, respectively.

**Table 5 animals-10-01539-t005:** Ammonia concentrations in manure headspace.

NH_3_ Concentrations [mg × m^−3^] (*n* = 5)						
C	BC2	BC4	BCM2	BCM4	SEM	*p*-Value
7.29 ^a^	6.20 ^b^	6.25 ^b^	6.05 ^b^	6.22 ^b^	0.13	0.00115
PRC (%)	−15	−14	−17	−15		

C = control group; BC2 = 2% of beechwood biochar by mass; BC4 = 4% of beechwood biochar by mass; BCM2 = 3% of beechwood biochar-based additive by mass; BCM4 = 6% beechwood biochar-based additive by mass. The beechwood biochar mass in the BCM2 and BCM4 groups was identical to BC2 and BC4 groups, respectively. ^a,b^ Mean values in the same row marked with different superscript indicate statistical significance (*p* < 0.05). PRC = percentage in relation to the control. Negative PRC indicates a reduction, while positive PRC values indicate an increase, in comparison to the control group.

**Table 6 animals-10-01539-t006:** Odor concentrations in manure headspace.

Odor Concentrations (OU_E_ × m^−3^) (*n* = 3)				
C	BC4	BCM4	SEM	*p*-Value
704.33	476.00	444.66	59.43	0.0972
PRC (%)	−32	−37		

C = control group; BC4 = 4% of beechwood biochar by mass; BCM4 = 6% beechwood biochar-based additive by mass. The beechwood biochar mass in the BCM4 group was numerically the same as in the corresponding BC4 group. PRC = percentage in relation to the control. Negative PRC values indicate a reduction in comparison to the control group.

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
