# Peer review of "Effect of Biochar Diet Supplementation on Chicken Broilers Performance, NH3 and Odor Emissions and Meat Consumer Acceptance"

_animals, 2020, doi:10.3390/ani10091539_

Round 1
Reviewer 1 Report
Dear authors,
It is good that you have accepted the comments and made adjustments. The quality of the article has now improved, so I can recommend to print the manuscript in the journal
Author Response
Dear reviewer, we are pleased that the revisions we have made are satisfying. Thank you for your feedback that helped improve the manuscript.
Reviewer 2 Report
Some my concerns has not been well addressed, I suggest the paper should be major revised.
- Table 3: Although some mean values are not significantly different from any other values, the superscript of "a,b" still be added. Like the ABW of week 5, and the FCR.
- Again, the data of the average daily feed intake sould be added. If there is no the date of ADFI, How the FCR can be calculated. I suspect the result of FCR is incorrect.
Author Response
Dear reviewer, please find our response in the attached .docx file.

Reviewer 3 Report
Dear Authors,
I have no comments.
Author Response

(The authors gave the same response as above.)

Round 2
Reviewer 2 Report
Table 2: Again, the superscript of "a,b" still need to be added for some values, for example of the ABW of week 1, it should mark like this: 148.33a 140.58ab 138.33ab 142.98ab 135.12b. You should mark all the values in the same row.
The average daily feed intake is a very important production parameter. I do not understand why the author can not add it. It is very easy to calculate. ADFI=AMF/the duration of the experiment (day).
Author Response
Dear reviewer, please find our response in the attached .docx file, written in the blue font.

This manuscript is a resubmission of an earlier submission. The following is a list of the peer review reports and author responses from that submission.
Round 1
Reviewer 1 Report
- Simple Summary: lines 19-20 “the reduction of odor ..... daily weight were lower” and in Abstract: lines 29-30: “Both feed additives showed a statistically significant reduction of ammonia and odor emissions by up to 15% and 37%, respectively "- incorrect statements.
As the study data show, only ammonia emissions decreased significantly, and the decrease in odor concentration was not statistically significant.
- lines 19-20 reads “The results show that .... increased daily feed intake.“
However, feed intake is not presented and analyzed in the „Results“, so it is not necessary to write about it in the abstract and state that it has increased.
- Lines 107, 169, 190, 219, 228, 243 - incorrectly indicated numbers of analyzed tables and figure.
- The conclusions are not written correctly.
I propose to review the conclusions and clarify them. The conclusions must be precise, clear and concise. The following expressions should be avoided: line 324- 332 "In general ....... than used in the present study" - Such statements are more relevant to the Discussion section than to the Conclusions.
Author Response
Dear Reviewer, thank you for you comments. Please find our responses in the attached file, written in the blue text.

Reviewer 2 Report
The aim of the present study aimed to evaluate the effect of biochar diet supplementation for broiler chickens on ammonia & odor emissions from manure, feed conversion ratio & daily weight gain and selected meat quality & sensory parameters. This is an important contribution that will be of interest to agricultural scientists, the agriculture industry, feed manufacturers, nutritionists, regulators, and the supplement industry. However, some concerns to be addressed as follows:
- Line 27: You cannot start a sentence with the number.
- Lines 35-36: As there is no significant difference in carcass mass, please revise the sentence.
- Line 71: The full name of abbreviations should be written for the first time.
- Line 99: What is the main chemical composition for BC? Please add more detailed information.
- Line 181, Table 3: Some superscript of “a,b” were missing, it should be added. Why there is no data on the average daily feed intake? Please add it. d
- Line 318-336: The conclusion is not suitable. Please shorten the conclusion and focus on the most important aspects.
Author Response

(The authors gave the same response as above.)

Reviewer 3 Report
This is an interesting study, very well prepared and very well written. I checked both for scientific and language content, and I am happy to say that no mistakes were found.
Author Response

(The authors gave the same response as above.)

Reviewer 4 Report
Dear authors,
you provide interesting study. Intensive poultry production raises concerns about gaseous emissions and its impact on workers, birds, and the environment. In this aspect, undertaking this research topic is current and necessary.
Please use the comments to improve the manuscript. All comments were placed in the text.

Author Response

(The authors gave the same response as above.)
